# Toward Management of Uncertainty in Self-Adaptive Software Systems: IoT Case Study

**Shereen Ismail \*** **, Kruti Shah** , **Hassan Reza** , **Ronald Marsh and Emanuel Grant**

School of Electrical Engineering and Computer Science, University of North Dakota,
Grand Forks, ND 58201, USA; kruti.shah@ndus.edu (K.S.); hassan.reza@ndus.edu (H.R.);
ronald.marsh@ndus.edu (R.M.); emanuel.grant@ndus.edu (E.G.)
**\*** Correspondence: shereen.ismail@ndus.edu

**Abstract:** Adaptivity is the ability of the system to change its behavior whenever it does not achieve the system requirements. Self-adaptive software systems (SASS) are considered a milestone in software development in many modern complex scientific and engineering fields. Employing self-adaptation into a system can accomplish better functionality or performance; however, it may lead to unexpected system behavior and consequently to uncertainty. The uncertainty that results from using SASS needs to be tackled from different perspectives. The Internet of Things (IoT) that utilizes the attributes of SASS presents great development opportunities. Because IoT is a relatively new domain, it carries a high level of uncertainty. The goal of this work is to highlight more details about self-adaptivity in software systems, describe all possible sources of uncertainty, and illustrate its effect on the ability of the system to fulfill its objectives. We provide a survey of state-of-the-art approaches coping with uncertainty in SASS and discuss their performance. We classify the different sources of uncertainty based on their location and nature in SASS. Moreover, we present IoT as a case study to define uncertainty at different layers of the IoT stack. We use this case study to identify the sources of uncertainty, categorize the sources according to IoT stack layers, demonstrate the effect of uncertainty on the ability of the system to fulfill its objectives, and discuss the state-of-the-art approaches to mitigate the sources of uncertainty. We conclude with a set of challenges that provide a guide for future study.

**Keywords:** self-adaptive software systems; uncertainty; sources; solutions; Internet of Things

## 1. Introduction

System adaptivity has been a topic of research since the mid-1960s. Adaptive software systems are capable of evaluating and changing their behavior whenever the system evaluation shows that the software is not accomplishing what it was intended to do, or when better functionality or performance may be possible while keeping system complexity hidden from the user. Adaptivity is the capability of the system to adjust its behavior in response to internal and external effects [1]. These causes can be categorized into internal and external. Internal causes may include variations in sensor readings, not achieving the predetermined objectives, system infrastructure properties, variations in system resources (to proactively) manage a changing system load during certain anticipated situations), or internal system faults (hardware defects or service failures). External causes may include disturbances in the environment (effects from external sources of noise or signal interference), adding or removing system requirements or changes in the priority of those requirements based on stakeholders needs [2], or any other unexpected events such as a sudden increase in the number of users' requests [3]. Therefore, adaptivity can, and should, be evaluated at design time, during the development of the system, and after deployment when the system is operating and continuously monitored.

Adaptive software systems can be classified into two categories based on the expected impact on the system and/or cost factors (i.e., runtime, development time, system resources,

and complexity)—weak adaption and strong adaption [3]. Weak adaptation employs low-cost impacts such as modifying parameters or other actions such as changing the transmitted data type or compressing data before transmission. Strong adaptation employs high cost or extensive impact actions such as changing the underlying system architecture, adding, or removing artifacts, replacing system components, resource provisioning, or even restarting/re-deployment.

In mono-model adaptive systems, a single model is supported by the system and the term adaptation mainly depends on the user requirements, system properties, and environmental characteristics. During runtime, design parameters or relations are modified to achieve the perceived behavior. In multi-model adaptive systems, where the system dynamically supports several models, each one can still satisfy the system objectives, but, depending on the model, with different impacts on some non-functional quality attributes. In such systems, adaptivity can be defined as the capability of a system to achieve its objectives by selectively switching execution between these models [4].

In a self-adaptive software system (SASS), the "self" prefix indicates that the system autonomously decides how to adapt or to organize so that it can accommodate changes in its contexts and operating environments [5]. Self-adaptive is sometimes paraphrased in the literature as self-managing, self-healing, self-optimizing, self-configuring, or self-organizing. A self-adaptive system consists of a closed-loop system [3] (i.e., modifying itself through the feedback of internal connections due to continuous changes during system runtime), thereby minimizing human efforts in the computer interaction [6].

SASS is considered a milestone in software development in many modern complex scientific and engineering fields [7] such as high-performance computing, control systems, programming languages, fault-tolerant computing, biological computing, and natural systems, embedded systems (i.e., mobile and autonomous robotics), machine learning, economic and financial systems, business and military strategic planning, Internet of Things (IoT), etc. [8,9]. These modern systems that are performing multiple tasks in diverse scenarios have to make decisions on adaptivity at runtime concerning changing requirements and dynamic operating environment.

The idea behind SASS can be represented by a feedback loop, as Figure 1 shows. The feedback loop provides the generic mechanism for self-adaptation. It collects pertinent knowledge at runtime, monitors the system functionality, and applies changes to subsystems when necessary, regardless of possible uncertainties [7].

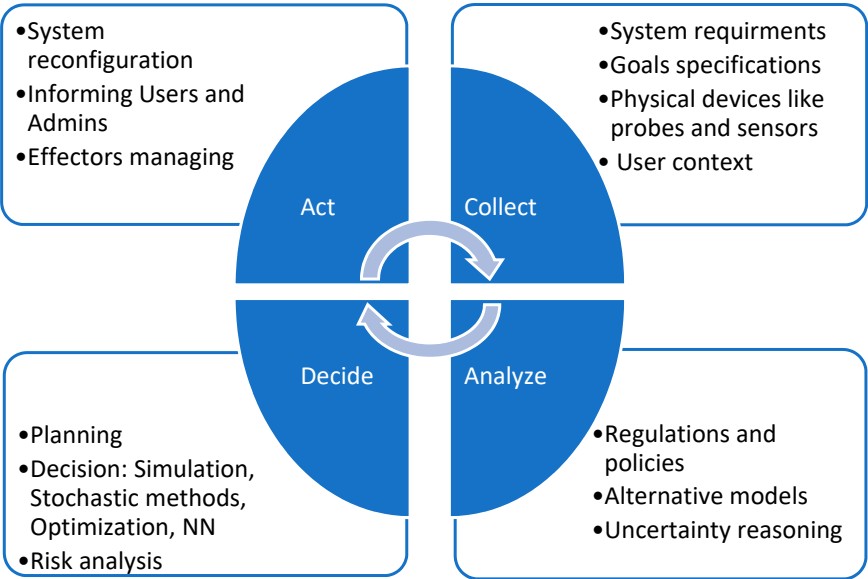

**Figure 1.** Self-adaptive software system (SASS) in feedback loop [7].

As shown in Figure 1, a feedback loop typically consists of the following steps: collect, analyze, decide, and act. A feedback cycle starts with collecting raw data from sources such as system requirements, goals specifications, physical devices, and user context. Then, the collected data are analyzed by following system regulations and policies, alternative models, and uncertainty reasoning. After that, a decision is made regarding how to adapt the system to reach the desirable state by planning, evaluating the decision (based on modeling the system using simulation, stochastic methods, optimization, or neural network), and risk analysis. Finally, the system acts to implement the decision to drive the system into the targeted direction through system reconfiguration, informing users along with administrators and managing effectors [7].

Self-adaptive systems at normal operation need to achieve multiple non-functional requirements based on stakeholder needs. Behavior may deviate from the expected one due to various runtime dynamics and events that are difficult to predict at design time [2]. With a growing number of both adaptation scenarios and requirements trade-offs that require the system to prioritize certain adaptation actions, choose the optimal adaptation scenario, adapt the system, and handle the positive or negative chain of effects caused by the adaptation of certain requirements [10,11]. Consequently, when the number of system non-functional requirements increases, so does the number of adaptation alternatives, and handling the requirements trade-offs becomes more complex. This issue should be investigated properly; otherwise, uncertainty may induce inconsistency in certain subsystems, and the accumulated inconsistencies may result in unignorable circumstances that adversely affect system behavior.

In this paper, we focus on the IoT field because it is a long-range technological area that presents great opportunities for development. Because IoT is essential infrastructure for vital applications that utilizes the attributes of SASS, it carries a high level of uncertainty, both in relation to IoT technologies and to the aspects that are (or will be) correlated to this field, such as social, economic, technological, legal, etc. IoT applications are subject to a variety of uncertainties such as heterogeneity of edge nodes, scalability, transmission technology, optimum energy use, user configuration capabilities, data management, reliability, privacy protection, security, and legal regulatory compliance [9]. IoT field is a long-range technological area that utilizes the attributes of SASS, which presents great opportunities for development. There is a great potential for applying IoT technology across emerging sectors including both industrial and public to improve uncertainty. We are looking deeply into causes of uncertainty and quality attributes that may affect uncertainty in this system, and discuss the possible solutions [12].

The contributions of this work are as follows. First, we define uncertainty using a high-level view of SASS and describe what are the sources of uncertainty in this system, which provides a common terminology for looking at the problems and communicating ideas. The goal of this work is to classify the different sources of uncertainty based on their location and nature in SASS. Second, we consider IoT as a case study of SASS. To take an advantage of self-adaption, this study addresses the phenomenon of uncertainty occurring in large, developing systems namely IoT through the example of various IoT application systems. In addition, we categorize the sources of uncertainty according to IoT stack layers.

The remainder of this paper is structured as follows. Section 2 defines and classifies uncertainty in SASS. Section 3 illustrates a high-level model that defines uncertainty representation in SASS. Section 4 discusses the sources that need to be considered to mitigate uncertainty. Characteristics of uncertainty in SASS are explained in Section 5. Section 6 presents the mathematical techniques for representing uncertainty. In Section 7, we summarize the state-of-the-art prominent approaches that tackle uncertainty in SASS. Section 8 defines uncertainty in IoT as a case study of SASS, which presents great opportunities for development. We illustrate the main layers of an IoT stack and discuss the sources which need to be considered to mitigate uncertainty. Moreover, we summarize the possible solutions to tackle uncertainty in IoT. We conclude our paper with future works in Section 9.

## 2. Scope of Uncertainty in SASS

In a software system, uncertainty is defined as the conditions in which the behavior of the system deviates from expectations due to the dynamicity and volatility of a variety of factors. One way to deal with this uncertainty is to build systems that, when information is available, adapt during runtime. However, the integration of self-adaptation into a system may lead to more uncertainty because faulty adaptation acts or unintended effects of system adaptation can lead to unexpected system activity [13].

The essence of a SASS is making the adaptation decision. Uncertainty arises due to the deficiency of adequate knowledge or experience to make correct decisions about adaptation. The adaptation decision can be made with conviction when all the information needed to make these choices is completely available. However, it is difficult to provide the correct amount of required information at the right time [14].

In general, uncertainty is known as a second-order notion in the field of software engineering [15]. A conventional misbelief is that the impact of uncertainty can be eliminated by a series of procedures to allow concentrating on standard behavior. Even though it is usually true that providing more facts reduces the measure of uncertainty, it is normally not possible to remove uncertainty entirely (because gathering all the data about a system is not feasible or achievable); hence, the degree of uncertainty can be reduced [16].

As sources of uncertainty, we refer to the factors such as users' frequent struggle to indicate their quality expectations correctly at design time, deployed sensors readings having unmanageable noise, simplified assumptions use in analytical models to determine the quality attributes of the software, etc. These elements question the belief with which the choices for adaptation are formed. We believe that treating uncertainty as a first-class concept increases the consistency or the accuracy of adaptation decisions [16].

Some sources of uncertainty can be categorized into internal and external. Internal uncertainty is caused by the effect of adaptation in a system with multiple objectives and quality goals making the process for selecting the optimal adaptation action is quite complex or the effect of modifying or substituting a software part of the system [17]. For example, in the network Open Systems Interconnection model (OSI) model, the situation where the system modifies the transport layer protocol from Transmission Control Protocol (TCP) to User Datagram Protocol (UDP) can be considered as an internal cause of uncertainty because of the differences in the way of manipulating data transmission that may affect the whole network progress. On the contrary, external uncertainty induces by the dynamicity or complexity of the environment. For instance, in certain weather conditions such as a snowstorm, an autonomous vehicle navigator may induce external uncertainty into the system. If the navigator part of the vehicle is substituted by a more stable or conventional navigator that can predict these types of weather conditions, it will help to prevent any possible crash [17]. Approaches for modeling various kinds of uncertainties are very different from each other as we will mention later in Section 7 [16].

## 3. Modeling Uncertainty in SASS

A high-level view of SASS is described in Figure 2 [18]. The SASS is split into two sections in this model— meta-level and base-level. The key functionality of the program is provided by the base-level subsystem. Meta-level subsystem leads the base-level subsystem via reflection on its behavior. A feedback control loop according to MAPE-K (Monitor-Analyze-Plan-Execute over a shared Knowledge) reference architecture is within the meta-level subsystem [14].

User and Environment are the other two entities of a self-adaptive software system as shown in Figure 2. The user objectives will be stated and passed from the base-level subsystem to the meta-level subsystem. The self-adaptive software system resides in an environment and thus continuously communicates with elements of that environment at the base-level subsystem because it is the duty of the meta-level subsystem to satisfy the user objectives stated at the base-level subsystem and monitor the environment.

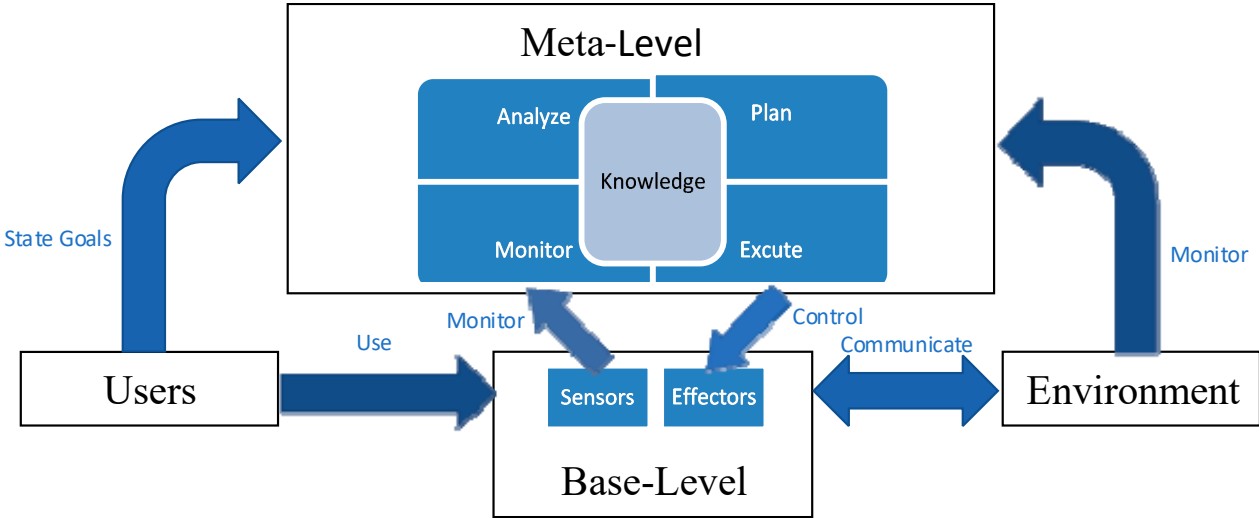

**Figure 2.** The high-level view of a self-adaptive software system model.

The adaptation choice made by the meta-level subsystem requires modeling all other entities that appear in Figure 2. The entities are loosely coupled in the self-adaptive software system. The interfaces of the metal-level subsystem with the other components are one of the main places in which uncertainty resides. We elaborate on the various sources of uncertainty faced by a self-adaptive software system in Section 4.

This section may be divided by subheadings. It should provide a concise and precise description of the experimental results, their interpretation, and the experimental conclusions that can be drawn.

## 4. Sources of Uncertainty in SASS

One should consider the sources (factors) explained below in order to minimize uncertainty in SASS:

### 4.1. Uncertainty Due to Simplifying Assumptions

This source of uncertainty is connected to the "Control" interface in Figure 2 and is due to the inaccuracy of the complex base-level subsystem representing analytical models. These analytical models are used to describe the effect of adaptation choices on the quality attributes of the system. When modeling abstractions become an inaccurate reflection of the system, the error in those predictions is magnified. The fact that often the assumptions underlying the model are not kept at runtime is one of the reasons for uncertainty. For instance, an analytical model that quantifies the response time of the system will account for the dominant variables (such as component execution time and queuing time) and neglect others (such as the difference between TCP and UDP in transmission delay of packets). Response time estimates given by such a formulation vulnerable to errors.

### 4.2. Uncertainty Due to Model Drift

This source of uncertainty is also connected to the interface of "Control." The adaptation itself is another cause of uncertainty in the analytical models; the base-level subsystem does not implement adjustments exactly as demanded by the meta-level subsystem, which triggers a drift from the models. For instance, the situation where the meta-level subsystem asks the base-level subsystem to modify the transport layer protocol from TCP to UDP is considered in the same example. The base-level subsystem, however, fails to implement this shift, making the meta-level subsystem analytical models inconsistent with the base-level subsystem behavior.

### 4.3. Uncertainty Due to Noise in Sensor

This source of uncertainty refers to the interface of "Monitor" and is due to fluctuations in a circumstance, such as a system parameter that is monitored, which hardly refers to a single value, but rather to a series of values obtained during the time of observation. Consider that any time a sample is taken, a sensor measuring the available network bandwidth will return a slightly different amount, whereas the actual value of the bandwidth can be already set at the source. To show the error in the probes used, this form of uncertainty is referred to as noise.

### 4.4. Uncertainty Due to Human User in the Loop

This uncertainty is due to a paradigm shift from software types deployed on isolated servers to software types that involve human users in their daily activities. These new forms of software systems communicate intensively with human users and depend on appropriate human behavior. Human behavior, however, is inherently unpredictable, which consecutively induces uncertainty in the software system. The interface between the base-level subsystem and the user is associated with this form of uncertainty. For instance, in telemedicine, remote monitoring involves the communication of the user with the system to improve patient health status. However, as described above, the unpredictability of human behavior causes this kind of uncertainty.

### 4.5. Uncertainty in the Objectives

The "State Goals" interface refers to this form of uncertainty and is due to the complexity of the specifications and desires of the users. Multiple users with different perspectives may have several issues with the software system, some of which can conflict with one another. However, obtaining and reflecting the desires of the users in an accurate manner is a challenge. If stating goals can be made clear and consistent between all system users, this kind of uncertainty can be reduced.

### 4.6. Uncertainty Due to Decentralization

In a decentralized software system, information is distributed among all the system entities. No single entity has complete information of all other entities states in the decentralization system. These entities can only access other entities' interfaces. All these entities need to co-ordinate their work and obtain information about other entities in order to achieve the system goals and reduce this uncertainty.

### 4.7. Uncertainty in the Execution Context

The meta-level subsystem is required to recognize the shift in context and adjust the base-level subsystem to behave properly. As the context in which a system executes changes, the different entities in the system's environment also need to adapt to the change, failure in which introduces uncertainty.

Table 1, which is adopted from [19], represents the classification of uncertainty based on the following three categories: Context uncertainty is an identification of the model's boundaries, which is uncertainty about the data to be modeled. Regarding the real world, this uncertainty concerns the completeness of the model. Model structural uncertainty is concerned with the type of the model itself. This uncertainty relates to how closely the model's structure reflects the subset of the real world to be modeled. Uncertainty of input parameters is often known as the uncertainty of parameters and is correlated with the actual value of variables given to the model as input and with the methods used to calibrate the parameters of the model.

Based on their nature, Table 1 characterizes the sources of uncertainty due to aleatory (i.e., variability) or epistemic (i.e., lack of knowledge). The epistemic uncertainty is related to simplifying assumptions, model drift, human in the loop, objectives, and decentralization. The lack of complete knowledge makes adaptation decisions vulnerable to uncertainty. On the other hand, aleatory uncertainty is related to noise and context. In this case, uncer-

tainty is rooted in the fact that after the adaptation decision is made, the behavior of the system may change.

**Table 1.** Classification of different sources of uncertainty based on their location and nature.

| Source of Uncertainty | Classification Based on Location | | | Classification Based on Nature | |
|---|---|---|---|---|---|
| | **Structural** | **Context** | **Input Parameter** | **Aleatory** | **Epistemic** |
| Simplifying assumptions | ✓ | ✓ | | | ✓ |
| Model drift | ✓ | | | | ✓ |
| Noise in Sensing | | | ✓ | ✓ | |
| Human in the loop | | ✓ | | | ✓ |
| Objectives | | ✓ | ✓ | | ✓ |
| Decentralization | ✓ | ✓ | | | ✓ |
| Execution context | ✓ | ✓ | ✓ | ✓ | |

## 5. Characteristics of Uncertainty

We enumerate various characteristics of uncertainty in the following subsections.

### 5.1. Reducibility vs. Irreducibility

Irreducible uncertainty is associated with things that are implicitly unknown, whereas reducible uncertainty is associated with knowable items that are undetermined for that period. One of the key factors behind irreducible uncertainty is the unmanageable complexity of events. Internal uncertainty (e.g., the effect of modifying or substituting a software part of the system) can be considered as a reducible uncertainty. External uncertainty (e.g., the impact on the system due to change in weather conditions) can be considered as an irreducible uncertainty.

### 5.2. Variability vs. Lack of Knowledge

Uncertainty based on nature can be classified as aleatory or epistemic as shown in Table 1 [19]. Aleatory uncertainty is often due to variability and is usually modeled using probabilities. On the counterpart, Epistemic uncertainty refers to a lack of information and is often referred to as uncertainty of parameters. Techniques used to mitigate aleatory uncertainty can differ from methods used to mitigate epistemic [16].

## 6. Mathematical Techniques for Representing Uncertainty

In this section, we discuss two common mathematical techniques for describing uncertainty in SASS. Although uncertainty in SASS can be represented using both approaches, the ability to make adaptation decisions is significantly impacted by the representation.

### 6.1. Probability Theory

The probability theory [20] is the most common approach to handle uncertainty (managing unknown circumstances) in SASS [14]. There are two well-known probability evaluations on fundamental differences—classical interpretation theory and Bayesian interpretation theory [20]. The theory of probability had previously been expressed by the classical interpretation of the theory. Based on its central premise, the classical interpretation outcome of a phenomenon is equally possible. Interpretation of this theory creates inconsistencies when it is applied beyond games of chance. Due to this limitation, the frequentist interpretation was designed. In this interpretation, the probability of an event in a large number of trials is specified as the limit of its relative frequency. Bayesian

theory [21] is defined on the basis of selective interpretation of probability theory. The probability in this interpretation is characterized as a representation of a human's rational belief in uncertain propositions. The frequentist interpretation is extended by Bayesian theory because it enables probability assignment to a single observation regardless of whether it is part of a larger observation or not. In cases when there are not sufficient data for frequentists, this interpretation is very beneficial. For example, a new unknown entity for which adequate data are not available cannot be evaluated by frequentist interpretation, whereas Bayesians can use subjective knowledge based on the specific phenomenon to analyze the new concept. In the case of both full and partial availability of the information, Bayesian theory is a single approach that can be applied.

### 6.2. Fuzzy Sets and Possibility Theory

An extension of the classical theory is the fuzzy set theory [22]. In classical set theory, if an element is not a member of the set (for unknown quantity), value 0 is assigned, while 1 is assigned if the element is a member of the set. According to the fuzzy set theory, any value between 0 and 1 is the membership value of an element with respect to a set. The higher the degree of membership, the more likely the element belongs to the set. Accordingly, there is no specified definition for the boundary of a fuzzy set, while the boundary of a classical set is defined by a crisp value (0 or 1). In systems where data are inaccurate or unclear, the fuzzy set theory is useful. For instance, in the anti-lock braking system, fuzzy logic is used to controls brakes in hazardous cases depend on car speed, acceleration, wheel speed, and acceleration.

Although probability theory is concerned with evaluating numeric value due to the change in data, possibility theory concentrates on measuring the information ambiguity. Nevertheless, the two theories (probability and possibility) can be used interchangeably. In general, the principle of possibility is useful when little knowledge or inaccurate data are available. However, probability theory is used if detailed knowledge is available.

## 7. Approaches for Handling Uncertainty in SASS

There is a lack of relevant strategies in this context for dealing with uncertainty. In this section, we will briefly present the approaches that minimize uncertainty in SASS.

### 7.1. Rainbow

In the Rainbow [23] framework, high-quality strategies are used to reduce uncertainty in SASS architecture. Rainbow focuses on sources of uncertainty correlated to MAPE–K feedback control loop [23], which are (1) identify the system faults (monitor and analyze phase), (2) select the correct adaption strategy (decide and act phase), and (3) check adaption achieved in the system according to required goals ("plan" and "act" phase).

They use probability theory to calculate the running average (i.e., the average that will continue to change with new observations take place) in monitoring and address the fluctuation in the system environment to minimize uncertainty. Afterward, data are compared with stochastic details in SASS. If the issue is established, a mitigation strategy is then chosen to resolve it. To reduce the uncertainty in strategy selection, the Stitch model is used. Stitch makes it feasible to model the uncertainty in strategies directly. As a consequence, when a strategy is determined by the Rainbow system, it can decide it based on the expected value, which is a reflection of the underlying uncertainty. They determine the last source of uncertainty by deciding how long the framework can track the strategy's implementation before transferring the adjustment to the running system.

Rainbow mitigates the uncertainty by augmenting architectural models with probabilistic models due to simplifying assumptions and noise. In addition, Rainbow mitigates the uncertainty due to the drift in the architectural models by tracking the device after adaptation [14].

### 7.2. Possibilistic Self-Adaptation (POISED)

Possibilistic self-adaptation (POISED) [17] is a quantitative approach to handle the difficulties determined by making adaptation decisions automatically under internal uncertainty. Fuzzy set theory is used to estimate the uncertainty residing within the elements of SASS, which manages positive and negative effects of adaption decision. They suggested a new approach to enhance the quality(performance) and find a feasible solution for the best range of possible behaviors in reference to the utility of the system. To resolve the self-adaption problem, POISED describes possibilistic linear programming, which determines the trade-off between alternatives. In order to build the trade-off between distinct configuration alternatives, POISED relies on possibilistic linear programming in which they normalize and combine the objective function in order to specify priorities among the objectives. The POISED configuration allows the decision-maker to decide which element of uncertainty is more important, which targets the decision-making stage at runtime.

### 7.3. Anticipatory Dynamic Configuration (ADC)

Anticipatory dynamic configuration (ADC) [24] provides a facility (for dynamically changing resources) to choose suitable services to perform a task and assign resources at runtime between these services. ADC does not take environmental uncertainty into account. Due to this limitation (of a reactive approach), the authors expanded the original analysis in a subsequent study [25] to include anticipatory decisions and considered the inaccuracy of the expected use of resources. The authors of [26] enhanced the previous study by using modeling data to determine resource requirements of the system for various configurations. The anticipatory model selects a configuration that optimizes the cumulative expected value of utility over time by integrating resource availability prediction (using probability theory). This effectively decreases the number of modifications/configurations and system disturbances. The cost of switching between configurations is often considered for the adaptation decisions. Based on cost, ADC selects configuration. If the cost is low, ADP chooses a better configuration available at that time. On the contrary, a non-optimum configuration is chosen when the cost is high.

### 7.4. Feature-Oriented Self-Adaptation (FUSION)

Feature-oriented self-adaptation (FUSION) [27] is an online learning-based approach in SASS. Instead of relying on analytical models that are manually operated, which affect the system objectives based on adaption decisions, FUSION uses machine learning to adjust the system adaption behavior automatically in order to handle unexpected changes. The turnout of this learning consists of numerous relationships between the adaptation actions and the quality attributes of the SASS. The quality attributes can be measured by the runtime environment. The adaptation actions are related to fluctuation points in the software that can be applied at runtime.

FUSION involved two cycles—the learning cycle and the adaptation cycle. The learning cycle defines the relation between the quality attributes of SASS and adaptation actions. This cycle tracks the errors in the learned relation. The adaptation cycle gathers measurements and optimizes the system (using data collected from a runtime environment). If the quality factors (i.e., system quality attributes) reduce below a certain threshold point over time, the adaption cycle provides the gathered information and takes adaptation decisions accordingly to enhance the quality factors.

### 7.5. Resilient Situated Software System (RESIST)

Resilient situated software system (RESIST) [28] uses a feedback loop to monitor internal and external system properties, analyze the changes in system structure, and adapt the system configuration changes in order to improve reliability at runtime. The component-level reliability analyzer is used to predict the reliability of system components. RESIST focuses on mobile embedded widespread software. These software systems are extremely complex because there are challenges in the validation, verification, and testing

phases, which usually lead to uncertainty. This kind of system requires high reliability. RESIST can reduce the uncertainty with persistent (constant) learning. In addition, small variations in reliability are modeled as distributions of probability that signify noise.

RESIST uses a component-level reliability model that depends on dynamic learning to estimate the process that begins with analyzing the effect of the adaptation choices on the reliability of the system framework. The reliabilities of the components are measured stochastically using a discrete-time Markov chain (DTMC). Using probability theory mathematical technique, RESIST models the uncertainty in the learning process.

### 7.6. RELAX

RELAX [29] is a requirement specification language depends on fuzzy set theory that demonstrates environmental uncertainty in self-adaptive systems. RELAX defines the requirements that can be disabled or "relaxed" based on the state of the environment. Operators are designed specially to capture uncertainty, which can cause the relaxation of requirements). These operators also describe how the requirement can be relaxed at runtime.

RELAX language is expanded in a subsequent publication, [30] to specify flexible requirements within the goal model to handle uncertainty in the environment. They established a variety of threat modeling to identify the sources of uncertainty, which can cause a problem to satisfy system goals. The approach offers various tactics to reduce uncertainty, which are enabled by relaxing the requirements that can cause uncertainty.

In [31], the authors introduced a new approach that produces RELAXed goal specifications automatically. AutoRELAX identifies goals to RELAX by the specified operators and the shape of the fuzzy logic function that helps to reach the objective of the system. AutoRELAX creates solutions by designing tradeoffs between minimizing the number of RELAXed goals and maximizing functionality by reducing the number of adaptations caused by environmental factors.

### 7.7. FLAGS

To minimize the environmental uncertainty of the specified adaptive goals, FLAGS [32] uses fuzzy theory. Like RELAX, FLAGS enable the definition of tactics that must be taken if some goals are not satisfied. FLAGS aims to achieve the fundamental objective of adaptive systems at the level of requirements to minimize environmental uncertainty.

In addition to the software uncertainty, FLAGS also deals with the uncertainty in defining goals. FLAGS depends on two types of goals—fuzzy goals and clean (crisp) goals. Fuzzy goals are defined as requirements that are not completely defined. Linear temporal logic is used to specify the crisp goals and fuzzy temporal language is used to identify fuzzy goals some temporary errors are tolerated in case of defining imprecise objectives/goals.

## 8. Case Study: Dealing with Uncertainty in IoT

IoT is an example of SASS that carries a high level of uncertainty in various levels. IoT can be defined as a distributed network that consists of a large number of heterogeneous nodes that collect and exchange data and is deployed over a certain geographical area. IoT is recognized as one of the most important areas of future technology and is gaining vast attention from a wide range of industries. IoTs have many applications but all share the same challenges due to uncertainty. Uncertainty may be due to several causes such as dynamics during operation at different layers of IoT stack.

Figure 3 shows the main layers of the IoT stack. We highlight the dynamics that cause uncertainty according to these layers, especially if the system is applying self-adaption, which incurs additional costs in terms of energy and communication [33].

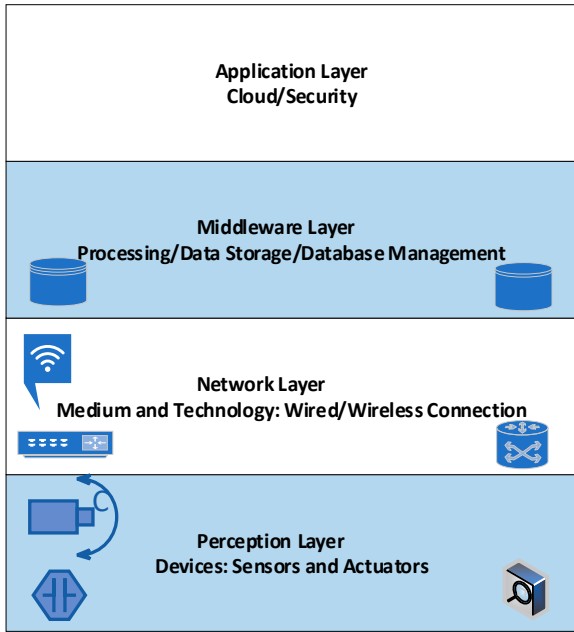

**Figure 3.** Main Layers of Internet of Things (IoT) stack.

At the perception layer, "things" may include any type of nodes, such as sensors, computers, or industrial equipment. For instance, different types of sensors are employed in IoT according to the application. It is well-known that sensor resources are limited in terms of energy, communication bandwidth, and processing capabilities. Uncertainty in sensor measurements may be due to sensor types, inaccurate sampling frequency, data compression, and amplification, or even possible hardware faults.

At the network layer, communication between "things" is subject to runtime uncertainties, such as interference that may affect ethernet cables in wired networks and noise, multi-path fading, or even mobility of devices that might affect the communication in wireless networks [34].

The middleware layer sits between applications and "things" and performs processing, filtration, and aggregation on collected data. Due to uncertainty at the middleware layer, a demand for in-network uncertain data management arises. For example, middleware uses application knowledge to perform data aggregation based on traffic priority. Aggregation reduces radio transmission and thus reduces energy consumption.

The application layer is responsible for formatting and representing data. The requirements of the application may be tuned at runtime, and adaption actions may require sensors to change the features of targeted data to fulfill the application goals and provide specific services to end-users.

With the rapid involvement of a high number of smart interconnected devices and sensors, IoT emerges in a wide range of technologies. IoT nodes communicate in a wireless ad hoc manner. Data integrations over different environments require to be supported by modular interoperable components. In addition, IoT infrastructure needs to combine volumes of data from various sources and use efficient lightweight aggregation techniques to better utilize the energy.

Heterogeneity of IoT widens the range of applications in different environments that require different networking technologies such as cellular, Wireless Local Area Network (WLAN), and Radio Frequency Identification (RFID). Communication technologies should be low cost with reliable connectivity and should apply privacy and security solutions that are lightweight and user appropriate.

In addition, legal regulations and standards are essential to allow all users to equally access and use system services and resources. Developments and coordination of legal

regulations and standards will promote the efficient development of IoT infrastructures and applications, services, and devices.

It is predicted that by 2025, there will be more than to 21 billion IoT devices. IoT devices such as machines and sensors are expected to generate 79.4 zettabytes of data in 2025, which is predicted by International Data Corporation (IDC). Excessive information transfer will cause greater uncertainty at received parties which affects making the correct decision.

IoT is considered a milestone in diverse domains applications such as manufacturing, e-health monitoring, smart home automation, smart cities, intelligent transportation, business and manufacturing, industrial automation, market, smart grid, smart safety, smart home, school, smart logistics, environment monitoring, surveillance systems, autonomous vehicles, smart agriculture and big data management, security surveillance, Wireless Sensor Network (WSN), etc. Due to concerns over size and cost, IoT motes typically offer limited computation, storage, and energy resources, which require careful design of IoT networks. However, with the new technological evolutions, more and more processing energy, storage and battery capacities become available at relatively low cost and low size that improve identification, communication, and processing capabilities. However, they add considerable uncertainty on various levels of processing, communication, and storage. The uncertainty is always evaluated at the beginning of software system development, but it must be a continuing process with solutions during the whole system's progress [35].

### 8.1. Sources of Uncertainty in IoT

Sources of uncertainty in IoT applications can be summarized as follows:

- Dynamics in the availability of resources (internal dynamics): Managing dynamics autonomously and correctly is especially important in highly critical IoT applications. For example, because the bandwidth of IoT terminals could vary from Kbps to Mbps from sensing simple value to video stream, requirements on hardware are diverging [12,34,36].
- Dynamics in the environment (external dynamics): These systems include unmanned underwater vehicles that are used for oceanic surveillance to monitor pollution levels, and supply chain systems that ensure sufficient, safe, and nutritious food to the global population. The dynamics of these systems introduce uncertainties that may be difficult or even impossible to anticipate before deployment. Hence, these systems need to resolve the uncertainties during system runtime [34].
- Heterogeneity of edge nodes: A large number of nodes may include nodes, sensors, actuators, and gateways, representing high diversity in their processing and communication capabilities. In IoT, one vital issue is integration and interoperability among these nodes (i.e., the ability to interconnect and communicate different systems) to form a cost-effective and easy-to-implement ad hoc network [37]. Devices connected through heterogeneous communications technologies such as Ethernet, Wi-Fi, Bluetooth, ZigBee, etc. consider various metrics, including the data range and rate, network size, Radio Frequency (RF) channels, bandwidth, and power consumption that needs to develop a heterogeneous technological approach to enable interoperable and secure communications in IoT. For instance, because the bandwidth of IoT terminals varies from Kbps to Mbps from sensing simple values to video streams, requirements on hardware are diverging [12].
- Scalability: IoT is an ad hoc network that formed quickly, and it changes rapidly where nodes are interconnected, using network services in a distributed manner [38]. Because the number of connected things is rapidly growing, IoT systems will require the composition of plenty of services into complex workflows. Accordingly, scalability in terms of the size of IoT nodes becomes a significant concern. Scalability will be measured by the capability of the system to handle increasing workloads, for instance, the addition or removal of IoT nodes or the addition or removal of computing resources in a single IoT node (e.g., adding more memory to increase buffer size or adding more processor capacity to speed up processing).

- Transmission technology: Wired and wireless technologies are the main transmission methods of data transfer [39]. Issues related to transmission are considered as an issue regarding transfer speeds and delays in the delivery of data [34]. For instance, in WSN, apart from uncertainty emerging from redundancy in densely deployed neighboring sensors, a significant amount of data can be lost or corrupted during the transmission from sensor nodes to the sink because of intrusion attacks, node failures, or battery depletion in sensor networks.

- Optimum energy use: Energy consumption is defined as the total energy consumed for running the components of the system architecture. The issue of power use is crucial [40]. The irregularities in energy consumption across the nodes, the size of battery capacity, and charging capability are challenging and require effective runtime decisions in IoT systems. The prediction of energy use under uncertainty is critical in fields such as WSN, e.g., forest monitoring or field surveillance systems. IoT nodes are deployed statically in the field to perform a certain task. A continuous source of energy is required to drive these nodes, which presents a severe challenge in terms of cost and network lifetime. Moreover, their computational and storage capabilities do not support complex operations [41].

- User configuration capabilities: The number of users, the complexity of systems, and the continuously changing needs of users make it necessary to provide mechanisms that allow the users to configure the systems by themselves. For instance, IoT in smart homes or smart buildings, including servomotors, mobile devices, televisions, thermostats, energy meters, lighting control systems, music streaming, and control systems, remote video streaming boxes, pool systems, and irrigation systems, in addition to various types of built-in sensors in equipment, and objects whose primary profiles did not include ICT functions (dishwashers, microwave ovens, refrigerators, doors, walls, furniture, windows, facades, elevators, ventilation modules, heating/cooling modules, water systems, roofs, electrical power systems, communication systems, office equipment, data storage systems, video monitoring and property control systems, and home appliances), which require young and elderly people to be able to cope with and sett configurations to meet their requirements [35].

- Data management: In IoT, it is crucial to utilize appropriate data models and semantic descriptions of their content, appropriate language, and format. For instance, when big data are processed and managed, the uncertainty causes severe liabilities with respect to the effectiveness and the accuracy of the big data [42]. To give an example, consider continuous measuring of temperature as a function of an environmental sensor network that produces big data streams. Due to readings, and possible transmission errors, it happens that temperature readings arrive at the receiver node with some uncertainty that is modeled in terms of probability distribution functions. Another example considers the function of a smart city application that determines geographical coordinates. Hence, the collection of geographical coordinates represents big data that are defined in hierarchical models on top of the geography of the smart city, such as stores, streets, quarters, districts, areas, cities, etc. Missing or processing errors causes smart data to be processed without an exact association to the hierarchical level of the smart city they refer to (e.g., the quarter where a certain store is located is known but the street of this store is not known).

- Reliability: The data loss may occur due to the inability of the system to maintain errors. Many factors can cause errors that decrease reliability and safety in IoT nodes, namely, (1) data distortion and corruption, so data can be changed due to imperfect software and/or hardware during data processing, transmission, and storage and (2) complete and partial data loss where the probability of technological disasters, virus attacks, human-made errors, etc. will lead to a complete or partial loss of data [9].

- Privacy protection: IoT nodes must ensure an appropriate level of security and privacy because of their close relationship with the real world [43]. For example, consider the case of obtaining data by unauthorized users, uncertainty in authentication stems

from the incompleteness of information regarding the acceptance of an authentication request leads to an incident. Moreover, uncertainty arises from a lack of precision (ambiguity) in the information requesting to authenticate in the system. For instance, assume that X attempts to authenticate to a system. We also assume that authenticating him endangers the system (the access would expose an asset to a threat) with a probability of 50%. A formal definition for uncertainty in authentication is presented as follows: given a set of authentication requests: R = {r1, r2, r3, . . . , rn}, a set of possible access decisions: D = {Access, Deny}, an access decision function: F:R→D, and a set of possible outcomes for any access decision: O = {Safe, Incident}; the uncertainty of an authentication request is defined by the probability.

- Security: Security in IoT ensures controlling access and authorizing legitimate uses only. Challenges and approaches are proposed to overcome the security issues in different layers of IoT [9]. For instance, in Application Programming Interface (API) malfunctions, clouds provide a set of User Interfaces (UIs) and APIs to customers to manage and interact with cloud services. The security and availability of services depend on the security of these APIs. Because they are used frequently, they are very likely to be attacked continuously. Therefore, adequate protection from the attacks is necessary. These interfaces must protect against accidental and malicious attempts to prevent security policies. Poorly designed, broken, exposed, and hacked APIs could lead to data ambiguity.

- Legal regulatory compliance: Governance strategies are a useful mechanism to address issues related to risk mitigation, compliance, and legal requirements [44]. However, supporting these strategies face challenges mostly due to uncertainties inherited in IoT infrastructure such as requiring changes in sensor configurations (i.e., transfer rate) or changing nodes communication protocol in situations such as an emergency or multiple devices failure that may lead to performance variability.

- Limitations of sensors in IoT: Due to limitations of sensors and uncertain measures, sensors readings require preprocessing because they induce uncertainty due to (1) missing readings (tag collisions, tag detuning, metal/liquid effect, tag misalignment); (2) inconsistent data: RFID tags can be read using various readers at the same time; therefore, it is possible to obtain inconsistent data about the exact location of tags; (3) ghost data: sometimes radio frequencies might cause data to be reflected in reading areas, so RFID readers might read those reflections; (4) redundant data: captured data may contain significant amounts of additional information; and (5) incomplete data: tagged objects might be stolen or forged and generate fake data [34].

*8.2. Classification of Sources of Uncertainty Based on Layers of IoT Stack*

We study the stack layers and all possible sources of uncertainty in IoT. We categorize them based on our understanding, nature of uncertainty, and system behavior toward different sources of uncertainty. Table 2 shows the classification of uncertainty based on IoT layers (i.e., perception layer, network layer, middleware layer, and application layer).

Dynamics in the availability of resources, which is considered an internal type of dynamics, and heterogeneity of edge nodes can be found at the perception and network layers. On the contrary, dynamics in the environment that are considered an external type of dynamics can be found at the perception and application layers.

Uncertainty due to scalability and reliability basically exists at the perception and middleware layers. Types of transmission technologies (wired/wireless) belong to the network layer. Optimum energy use and limitations of sensors are two causes of uncertainty at the perception layer. User configuration capabilities and legal regulatory compliance induce uncertainty at the application layer. Data management introduces uncertainty at the middleware layer. To mitigate uncertainty related to privacy protection, we need to focus on the network and application layers. Because security is related to end-users and data management, it needs to be considered at the middleware and application layers.

**Table 2.** Classification of different sources of uncertainty based on IoT Layers.

| Source of Uncertainty | Classification Based on Layer | | | |
|---|---|---|---|---|
| | Application Layer | Middleware Layer | Network Layer | Perception Layer |
| Dynamics in the availability of resources (internal Dynamics) | | | ✓ | ✓ |
| Dynamics in the environment (External Dynamics) | ✓ | | | ✓ |
| Heterogeneity of edge nodes | | | ✓ | ✓ |
| Scalability | | ✓ | | ✓ |
| Transmission Technology | | | ✓ | |
| Optimum energy use | | | | ✓ |
| User Configuration capabilities | ✓ | | | |
| Data management | | ✓ | | |
| Reliability | | ✓ | | ✓ |
| Privacy protection | ✓ | | ✓ | |
| Security | ✓ | ✓ | | |
| Legal regulatory compliance | ✓ | | | |
| Limitations of sensors | | | | ✓ |

*8.3. Proposed Solutions for Dealing with Uncertainty*

In this section, we discuss proposed solutions according to previously presented sources that can limit uncertainty in complex systems such as IoT.

- Heterogeneity of nodes: Selection of technology and types of nodes is an important factor that needs to be considered for IoT infrastructure. Acceptance of shared standards can be used to cope with the diversity of devices and applications. Adaptation of trust relationships needs to be implemented on the following levels in order to guide IoT devices to use the most trustworthy information for decision making and to reduce risk caused by malicious devices, i.e., IoT entities, data perception (sensor sensibility, preciseness, security, reliability, persistence, data collection efficiency), privacy preservation (user data and personal information), data aggregation, transmission, and communication, and human–computer interaction.
- Scalability: Service composition mechanisms are used to handle scalability requirements. A service composition mechanism defines a meaningful interaction between services by considering two functional dimensions—control flow and data flow. Control flow refers to the order in which interactions occur, and data flow defines how data are moved among services (behavior of workflow). With the increased number of devices, a data processing pipeline is required, which consists of a set of data and functions according to IoT applications that can appropriately be applied to the transferred data streams between various nodes. It is mandatory to have a system that can be easily expanded according to future needs [38].
- Reliability: Several mechanisms are used to improve reliability, minimize risks of data loss, distortion, and security violation. Ensuring that information generated by IoT is precise, authentic, up to date and complete is very important. Comprehensive approaches are used to detect possible errors in the design phase of the system to avoid uncertainty due to partial information. Replication is another solution that

helps to ensure data reliability but requires applying energy-efficient cryptographic algorithms, error correction codes, access structures, secret sharing schemes, etc. However, applying these techniques may increase data storage cost, load, processing for encryption and decryption, the transmission of secret keys, etc. [9].

- Privacy protection: We discussed possibility and probability theory earlier. Probability theory was chosen to define and handle uncertainty in authentication due to the challenges of IoT scenarios such as scalability and the need for less complexity. Privacy protection can be handled by "trust" analysis methods in order to design lightweight security protocols and efficient cryptographic algorithms. Prediction models can also be used in trust analysis. Supervised machine learning algorithms (classification algorithms) may be applied to the dataset in order to build prediction models. For instance, authentication requests were classified into "Access" or "Deny" class. Afterward, a behavioral-based analysis algorithm is used to check the history profile of the user to improve the accuracy of prediction models [43].
- Simulation and modeling uncertainty in IoT: The creation of simulations and models of uncertainty phenomena is a big challenge in IoT [33]. Numerical or statistical model checking at runtime will help select the proper configuration that complies with self-adaption and manage uncertainty in different IoT system components to evaluate system properties. For example, noise uncertainty will degrade the performance severely; in [45], the authors proposed a noise detector in IoT systems, which opens the door for others to find techniques that improve the power control and sensing accuracy in IoT devices.
- Legal regulation and standardization: Legal regulation, standardization, and security policies need to be implemented to overcome ethical and legal issues related to IoT. For example, in the data aggregation process, types of relevant standards are technology standards (including network protocols, communication protocols, and data-aggregation standards), and regulatory standards (related to security and privacy of data).
- Transmission technology: In wireless data transfer, a large sample size requires more energy, bandwidth, and latency. For instance, in WSN, traditional data compression techniques are not suitable, especially in dense networks. Avoiding oversampling beyond network capacity for a WSN is necessary to prevent excessive data transmission from sensors. Techniques that control transmission scheduling, queuing, and managing delay constraints need to be implemented for data transmission in IoT [39].
- Data management: In IoT, data management is handled by a layer between objects and devices that generate the data. Raw data or aggregated data will be transmitted via the network layer to data repositories. Users have access to these repositories to acquire the data [46].
- Optimum energy use: In most IoT applications, the energy consumption ratio is high. For instance, to reduce energy consumption, the adaption of active/sleep scheduling algorithms is employed to improve IoT lifetime. Other techniques for efficient energy consumption include low power radio, the use of self-sustaining sensors, network protocols that generate low traffic rates, and content caching [47].

## 9. Conclusions and Future Research

Employing self-adaption in software systems may lead to uncertainty. In this study, we presented a high-level model of SASS to study the conception and causes of uncertainty. We provided a survey of state-of-the-art approaches that tackle uncertainty from different perspectives. We classified the different sources of uncertainty based on their location and nature in SASS. The development of SASS faces significant challenges due to uncertainty since clear and consistent objectives are difficult to specify at design time and dynamic environment conditions are expected at runtime. Another challenge is quantifying uncertainty in SASS. If uncertainty can be even partially quantified/estimated, self-adaption in software systems is able to perform better than if it completely ignores uncertainty. Many

proposed approaches considered sources of uncertainty to a limited degree. To this end, we used IoT as a case study of SASS that is usually deployed in highly uncertain and rapidly changing environments. IoT is a critical infrastructure for many applications that utilizes the attributes of SASS. We studied the main layers of the IoT stack and categorized the sources of uncertainty accordingly. We then summarized the possible solutions to tackle uncertainty in layers of IoT stack.

One direction of future work is to apply a structured method that can handle multiple sources of uncertainty more efficiently. Our focus so far has been defining causes of uncertainty in the field of SASS. One of the limitations is that there is no optimized way to prioritize or rank uncertainty (i.e., high, medium, low) in SASS. As a part of future studies, researchers need to focus on developing algorithms and techniques that offer better detection and numerical representation of different sources of uncertainty in SASS.

**Author Contributions:** S.I., K.S. and H.R. identified the key architectural requirements and pattern such as uncertainly and MAPE-K used in CPS-based self-adaptive systems in this work. S.I. and K.S. carried out the experiments and quality analysis based on MAPE-K in Internet of Things based-on feedback from H.R., S.I. and K.S. wrote the paper. The paper was improved based on feedback and proofreading from R.M. and E.G. All authors have read and agreed to the published version of the manuscript.

**Funding:** This research received no external funding.

**Institutional Review Board Statement:** Not applicable.

**Informed Consent Statement:** Not applicable.

**Data Availability:** Not applicable.

**Conflicts of Interest:** There is no conflict of interest to declare.

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
