# Peer review of "Toward Management of Uncertainty in Self-Adaptive Software Systems: IoT Case Study"

_computers, doi:10.3390/computers10030027_

Round 1

Reviewer 1 Report

The paper deals with a very interesting and important issue. However, it should be improved substantially to increase its quality. With a view to increasing better understanding and development in the field of management of uncertainty in SAAS, it is worth stressing the own contribution made by the authors. So, please stress and describe all your results / own contribution in the proper way. Now it is no sufficient information concerning the own findings. The results of author’s own studies should be described as a separate section (part) of the paper to show the authors own contribution. The paper is based to a large extent on the results of other surveys and  extensive literature on the subject. In this perspective the provided description of state-of-the-art approaches (i.e. the possible sources of uncertainty, its effect on the ability of the system, approaches coping with uncertainty in SASS and discussion of their performance) has to be improved. The objective of these revisions should therefore focus on the demonstration of authors own contribution to the increasing research in management of uncertainty in SAAS issues.

The keywords are not good enough. Using twice the same word (adaptivity) is not necessary. The same situation applied to the term Self-Adaptive Software Systems and its abbreviation – SASS. Of course, the full term for which an abbreviation stands should precede its first use in the text. However, it is not necessary to repeat twice the same term as keywords.

The text, as a whole, appears clear and rightly technical in language. However, some formatting issues to be checked (i.e. in the Figures - points in titles are incorrect; literature sources are missing; page 3 at Line 105: “Since IoT is a relatively new domain…” – such statements are too generic, so please provide scientific justification; etc.).

Author Response

Dear Reviewer, 

I attached for you our response to your valuable comments.

Best,

Reviewer 2 Report

This article suggests a current and attractive topic for the academy. The research is timely and worthwhile. The research problem is clearly defined. The authors provide fresh insight into the field.

I hope you find the following observations helpful:

Structure: Articles should be reformatted according to a standard structure, which is set out in the instructions for authors of the journal (sections are Introduction, Materials and Methods, Results, and Discussions, Conclusion). See new template.

Figures 1 and 2 should be improved.

Results: Perhaps it is better to visualize in more charts. The result of this investigation is presented a high-level model of SASS. 

In my opinion, it may be better to provide the results of testing this model (if any) in the Results section.

Need to revise and check citations in the text and in the references section. I proposed you add references about Healthcare Monitoring.

Overall, I find the paper adequate but it can be improved by addressing the aforementioned issues. Especially the problem of the paper structure.

Congratulations on a job well done.

Author Response

(The authors gave the same response as above.)

Round 2

Reviewer 1 Report

Your changes are mostly presentational. However, the revised paper can be accepted for publication.

Reviewer 2 Report

The revised paper can be accepted for publication.

Possibly you will need to update your reference with a published article about healthcare monitoring software: Fedushko S., Ustyianovych T. Operational Intelligence Software Concepts for Continuous Healthcare Monitoring and Consolidated Data Storage Ecosystem. Advances in Intelligent Systems and Computing. 2021. (1247 AISC). C. 545–557.

Congratulations on a job well done.